# Lactoferrin Binding to SARS-CoV-2 Spike Glycoprotein Blocks Pseudoviral Entry and Relieves Iron Protein Dysregulation in Several In Vitro Models

**DOI:** 10.3390/pharmaceutics14102111

**Published:** 2022-10-03

**Authors:** Antimo Cutone, Luigi Rosa, Maria Carmela Bonaccorsi di Patti, Federico Iacovelli, Maria Pia Conte, Giusi Ianiro, Alice Romeo, Elena Campione, Luca Bianchi, Piera Valenti, Mattia Falconi, Giovanni Musci

**Affiliations:** 1Department of Biosciences and Territory, University of Molise, 86090 Pesche, Italy; 2Department of Public Health and Infectious Diseases, Sapienza University of Rome, 00185 Rome, Italy; 3Department of Biochemical Sciences, Sapienza University of Rome, 00185 Rome, Italy; 4Department of Biology, University of Rome “Tor Vergata”, 00133 Rome, Italy; 5Dermatology Unit, University of Rome “Tor Vergata”, 00133 Rome, Italy

**Keywords:** lactoferrin, SARS-CoV-2, COVID-19, iron homeostasis, inflammation

## Abstract

SARS-CoV-2 causes COVID-19, a predominantly pulmonary disease characterized by a burst of pro-inflammatory cytokines and an increase in free iron. The viral glycoprotein Spike mediates fusion to the host cell membrane, but its role as a virulence factor is largely unknown. Recently, the antiviral activity of lactoferrin against SARS-CoV-2 was demonstrated in vitro and shown to occur via binding to cell surface receptors, and its putative interaction with Spike was suggested by in silico analyses. We investigated the anti-SARS-CoV-2 activity of bovine and human lactoferrins in epithelial and macrophagic cells using a Spike-decorated pseudovirus. Lactoferrin inhibited pseudoviral fusion and counteracted the deleterious effects of Spike on iron and inflammatory homeostasis by restoring basal levels of iron-handling proteins and of proinflammatory cytokines IL-1β and IL-6. Using pull-down assays, we experimentally proved for the first time that lactoferrin binds to Spike, immediately suggesting a mechanism for the observed effects. The contribution of transferrin receptor 1 to Spike-mediated cell fusion was also experimentally demonstrated. In silico analyses showed that lactoferrin interacts with transferrin receptor 1, suggesting a multifaceted mechanism of action for lactoferrin. Our results give hope for the use of bovine lactoferrin, already available as a nutraceutical, as an adjuvant to standard therapies in COVID-19.

## 1. Introduction

Severe acute respiratory syndrome coronavirus 2 (SARS-CoV-2), the causative agent of coronavirus disease (COVID)-19, is an enveloped, positive-sense, single-stranded RNA betacoronavirus possessing about 79% identity to SARS-CoV [1]. SARS-CoV-2 has four major structural proteins, namely nucleocapsid, membrane, envelope and Spike [1]. The first critical step of viral infection is catalyzed by its trimeric Spike glycoproteins, which decorate the virion surface. Spike binds to angiotensin-converting enzyme 2 (ACE2) on the host cell through its receptor binding domain (RBD) within the S1 subunit, triggering proteolytic cleavage of Spike, fusion of the S2 subunit with the host cell membrane [2] and endocytosis of the viral particle [3]. In addition, the negative charges of cellular heparan sulfate proteoglycans (HSPGs) electrostatically interact with basic residues of Spike and strongly contribute to the early interaction between SARS-CoV-2 and host cells [2]. 

The pathology of SARS-CoV-2 is further worsened by the activation of innate immune cells and the release of inflammatory cytokines aimed at counteracting the viral infection. Activation of the immune response is essential for antiviral host defense, but an excessive release of proinflammatory cytokines, such as interleukin-6 (IL-6), could result in tissue injury, systemic inflammation, and organ failure [4]. In this regard, Spike is involved in the massive release of pro-inflammatory cytokines [5]. In particular, Spike (both from SARS-CoV and SARS-CoV-2) is a potent viral pathogen-associated molecular pattern (PAMP) sensed by toll-like receptor 2 (TLR2), which activates the NF-κB pathway, leading to the expression of inflammatory mediators in epithelial and innate immune cells [5]. However, the detailed mechanisms of the hyperinflammatory response during SARS-CoV-2 infection are still poorly understood.

Proinflammatory cytokines, particularly IL-6, can markedly influence iron homeostasis [6]. Since mammals do not present a direct iron excretion system [7], iron balance is strictly regulated by both enterocytes, through iron absorption, and macrophages, through iron recycling from senescent erythrocytes.

In humans, dietary iron is absorbed in the duodenum through the combined function of the ferrireductase duodenal cytochrome B (DCYTB) and the divalent metal transporter 1 (DMT-1) [8]. Once released in the cytoplasm, iron is exported into the plasma through ferroportin (Fpn), the only iron exporter identified in mammals so far [9]. To permit the final binding to serum transferrin (Tf), a ferroxidase is required [10]. This can be either hephaestin (Heph), mainly expressed by small intestine, or ceruloplasmin (Cp), synthesized by macrophages, hepatocytes and immune cells [10]. Via systemic circulation, Tf-bound ferric iron is then conveyed to sites of use/storage and released by receptor-mediated endocytosis. Uptaken iron is either promptly utilized by the cell or stored into cytosolic ferritin (Ftn) [10].

Iron homeostasis is grossly perturbed during infection and inflammation, leading to iron disorders. In particular, enterocytes and macrophages become iron-overloaded, thus increasing the susceptibility of the host to infections, including viral ones [11]. 

Within this framework, there has been in the last years a renewed interest in natural substances, such as lactoferrin (Lf), an iron-binding glycoprotein able to counteract viral infections and at the same time to rebalance iron and inflammatory homeostasis [12,13]. Lf belongs to the Tf family and is expressed upon induction by exocrine glands and neutrophils. As for other Tfs, its structure bears two lobes, lobe N and lobe C, each able to chelate one ferric ion [13]. However, at variance with other Tf members, Lf has peculiar physico-chemical features which make it a key factor in mammalian innate immunity. Lf chelates iron at very low pH, presents a marked cationic charge and can enter the cell nucleus, suggesting a role in modulation of gene expression. Such properties enable Lf to exert multifaceted functions, including antimicrobial, antiviral and anti-inflammatory activities [12,13]. Lf antiviral activity, demonstrated in several in vitro and in vivo models [11,12], largely relies on its ability to interact with both viral particles and host cell receptors, thus interfering with viral fusion to the host membrane [11,12]. However, other mechanisms are still under investigation. 

Most of the in vitro and in vivo studies, including antiviral ones, are carried out with bovine Lf (bLf), which shows about 70% sequence homology and identical functions to human Lf (hLf) [13]. Recently, a multimodal mechanism of action of bLf against SARS-CoV-2 infection has been proposed, either by its direct binding to host HSPGs [14] and to Spike on SARS-CoV-2 [15] or by modulation of the host cell innate immune response through increased expression of interferon-stimulated genes and tumor necrosis factor (TNF)-α [16]. 

Here, to validate the in silico results [15], where a direct recognition of the C-terminal domain 1 of the SARS-CoV-2 Spike glycoproteins by both bLf and hLf was described, we investigated the neutralizing activity of Lfs in different epithelial or macrophagic cell models using a pseudovirus decorated with the Spike protein. To experimentally demonstrate binding of Lfs to Spike, an in vitro pull-down assay was carried out. In addition, the effect of purified Spike on iron and inflammatory homeostasis in epithelial and macrophagic cell models, in the absence or presence of Lfs, was analyzed. Finally, the contribution of transferrin receptor 1 (TfR1) to Spike-mediated cell fusion was investigated and in silico approaches were applied to analyze Lfs interactions with TfR1 and the putative binding of bLf to different Spike glycoprotein variants.

## 2. Materials and Methods

### 2.1. Bovine and Human Lactoferrin

Highly purified bLf (Saputo Dairy, Southbank, Victoria, Australia) was generously supplied by Vivatis Pharma Italia s.r.l., and highly purified hLf was purchased from Sigma Aldrich (Milan, Italy). BLf and hLf purity was about 99% and 97%, respectively, as checked by SDS-PAGE and silver nitrate staining. The concentration of bLf and hLf solutions was assessed via UV spectroscopy with an extinction coefficient of 15.1 (280 nm, 1% solution). Iron saturation was about 11% and 9% for bLf and hLf, respectively, as determined via optical spectroscopy at 468 nm using an extinction coefficient of 0.54 for a 1% solution of 100% iron saturated protein. LPS contamination, assessed via Limulus Amebocyte assay (Pyrochrome kit, PBI International, Milan, Italy), was 0.5 ± 0.06 ng/mg for bLf and 0.3 ± 0.07 ng/mg for hLf. Before each in vitro assay, bLf and hLf solutions were sterilized using a 0.2 μm Millex HV filter at low protein retention (Millipore Corp., Bedford, MA, USA).

### 2.2. Cell Culture and Pseudovirus

The African green monkey kidney-derived Vero E6 and human colon carcinoma–derived Caco-2 cells were purchased from American Type Culture Collection (ATCC), Human Bronchial Epithelial (16HBE14o-) cell line was purchased from Millipore Sigma (St. Louis, MO, USA), while THP-1 cells were purchased from European Collection of Cell Cultures (ECACC). Vero E6 and Caco-2 cells were cultured in high-glucose Dulbecco’s Modified Eagle’s Medium (DMEM) (Euroclone, Milan, Italy) with 10% fetal bovine serum (FBS) (Euroclone, Italy) in a humidified incubator with 5% CO_2_ at 37 °C. 16HBE14o- Human Bronchial Epithelial cells were cultured in minimum essential medium (MEM) with 10% FBS at 37 °C in a humidified incubator with 5% CO_2_. THP-1 cells were maintained in RPMI 1640 medium (Euroclone, Italy), supplemented with 10% FBS and 2 mM glutamine, at 37 °C in a humidified incubator with 5% CO_2_. THP-1 cells, which grow spontaneously in loose suspension under these conditions, were subcultured twice a week by gentle shaking, followed by pelleting and reseeding at a density of approximately 5 × 10^5^ cells/mL.

SARS-CoV-2 Spike Pseudovirus (hereafter referred to as “Pseudovirus”), an HIV-based luciferase lentivirus pseudotyped with SARS-CoV-2 full length Spike protein of Wuhan strain, was purchased from Creative Biogene (New York, NY, USA) (SARS-CoV-2 S Pseudotyped Luciferase Lentivirus, cat. CoV-002). 

The Pseudovirus presents SARS-CoV-2 Spike as the only surface protein that mediates viral fusion with host cells.

### 2.3. Pseudovirus Neutralization Assay

For neutralization assays, cells were seeded in 96-well tissue culture plates (1 × 10^4^ cells/well) for 24 h (Vero E6) or 48 h (Caco-2 and 16HBE14o-) at 37 °C in a humidified incubator with 5% CO_2_. THP-1 cells were differentiated in macrophages by incubation in 96-well tissue culture plates at a density of 2 × 10^4^ cells/well in RPMI medium containing 0.16 μM phorbol myristate acetate (PMA) (Sigma Aldrich, Italy) for 48 h at 37 °C in a humidified incubator with 5% CO_2_. Cell confluence conditions were set following instructions provided by the Pseudovirus manufacturer. To evaluate the inhibition of Pseudovirus fusion to the host membrane, 1.25 and 6.25 μM of bLf or hLf, corresponding to 100 and 500 μg/mL, were used on Vero E6 cells; the higher concentration was used on 16HBE14o-, Caco-2 and THP-1 cells. For studies on the interaction of Lf with pseudoviral particles and/or host cells, the neutralization assay was carried out with a multiplicity of infection (MOI) of 10 in the presence or absence of bLf or hLf, according to the following experimental plan: (i) to evaluate the entry efficiency of the pseudoviral particles, cells were treated with Pseudovirus for 8 h at 37 °C; (ii) to evaluate whether Lf interferes with the viral fusion rate by binding viral surface components, the Pseudovirus was preincubated with bLf or hLf for 1 h at 37 °C and then the cells were treated with these suspensions for 8 h at 37 °C; (iii) to evaluate whether Lf interferes with viral attachment to host cells, cells were preincubated with bLf or hLf for 1 h at 37 °C. The cells were then washed with phosphate buffered saline (PBS) and treated with Pseudovirus for 8 h at 37 °C; (iv) to assess whether Lf interferes with both viral and host cell components, bLf or hLf was added together with Pseudovirus to the cell monolayer for 8 h at 37 °C. 

For experiments on the contribution of TfR1 to pseudoviral fusion to the cell membrane, two different approaches were followed: (i) cells were preincubated with an antibody against human TfR1 (sc-32272, Santa Cruz, CA, USA) for 1 h at 37 °C. The cells were then washed with phosphate buffered saline (PBS) and treated with Pseudovirus for 8 h at 37 °C; (ii) the Pseudovirus was preincubated with a soluble form of TfR1 (11020-H01H, Sino Biological, China) for 1 h at 37 °C and then the cells were treated with this suspension for 8 h at 37 °C.

At the end of the incubation, cells were washed twice with PBS, covered with the appropriate culture medium with 2% of FBS and incubated for 48 h at 37 °C in a humidified incubator with 5% CO_2_. After 48 h, cells were washed, lysed with cell culture lysis reagent (Promega, Italy) and the transduction efficiency was determined by luminescence analysis using firefly luciferase assay kit (Promega, Italy). The relative luciferase unit (RLU) in each well was detected using a Cytation 5 Cell Imaging Multi-Mode Reader (BioTek, Winooski, VT, USA).

### 2.4. Sepharose 6B Pull-Down

CNBr-activated Sepharose 6B (GE Healthcare, Chalfont St Giles, Buckinghamshire, UK) was employed for conjugation of bLf, hLf or human Tf (hTf, Fluka Sigma Aldrich, Milan, Italy). The resin (100 mg) was washed with 1 mM HCl and coupled to 0.5 mL of a 10 mg/mL protein solution in PBS by overnight incubation at room temperature under continuous shaking. The resin was fully inactivated by incubation in 1 mL of Tris-HCl 0.5 M pH 8.0 for 2 h at room temperature. After five washes with 1 mL of PBS, the resins were resuspended in an equal volume of PBS. An amount of 40 μL of the resuspended resins was added to 200 μL of full-length stabilized trimeric Spike of Wuhan strain (P2020-025, Trenzyme GmbH, Konstanz, Germany) (20 μg/mL) or its S1 domain (40591-V08H, Sino Biological, Eschborn, Germany) (20 μg/mL) and incubated for 2 h at room temperature under continuous shaking. The resins were then washed five times with 1 mL of PBS and eluted in 50 μL of SDS sample buffer. An amount of 20 μL of the eluted fractions was analyzed by SDS-PAGE and Western blot (monoclonal anti-His-HRP, Sigma, 1:10,000).

### 2.5. Stimulation of Caco-2 and Differentiated THP-1 Cells with Spike

For the stimulation assay, Caco-2 cells were seeded in 6-well tissue culture plates in complete DMEM medium at a density of 7 × 10^5^ cells/well for 48 h at 37 °C in a humidified incubator with 5% CO_2_, while THP-1 cells were differentiated in macrophages by incubation in 6-well tissue culture plates at a density of 2 × 10^6^ cells/well in complete RPMI medium containing 0.16 μM PMA for 48 h at 37 °C in a humidified incubator with 5% CO_2_. Caco-2 cells and differentiated THP-1 cells were washed twice with PBS and treated or not with full-length stabilized trimeric Spike and/or with bLf according to one of the following experimental procedures: (i) untreated cells; (ii) cells treated with 1.25 μM bLf; (iii) cells treated with 20 nM Spike; (iv) cells pre-treated with 20 nM Spike for 1 h and subsequent addition of 1.25 μM bLf; (v) cells pretreated with 1.25 μM bLf for 1 h and subsequent addition of 20 nM Spike and (vi) cells treated with a mixture of 1.25 μM bLf and 20 nM Spike preincubated for 1 h. For all conditions, cells were incubated for 48 h at 37 °C in a humidified incubator with 5% CO_2_.

After 48 h from treatments, cytokines were quantified on the supernatants. Adherent cells were scraped in PBS with 1 mM phenylmethylsulfonyl fluoride (PMSF), pelleted at 5000× *g* for 5 min and stored at −80 °C for protein analysis. 

### 2.6. Cytokine Analysis

Quantification of IL-1β and IL-6 was performed on cell monolayer supernatants using Human ELISA Max Deluxe Sets (BioLegend, San Diego, CA, USA).

### 2.7. Western Blots

Caco-2 cells and THP-1 cells were lysed in 300 μL of 25 mM MOPS pH 7.4, 150 mM NaCl, 1% Triton, 1 mM PMSF, 2 μM leupeptin and pepstatin in ice for 1 h. Total protein content was quantified by Bradford assay. An amount of 20 μg of total protein, in SDS sample buffer containing DTT, was heat-treated (except for Fpn [17]) and loaded onto SDS-PAGE. For Western blot analysis, the following primary antibodies were employed: monoclonal anti-TfR1 (anti-TfR) (sc-32272, Santa Cruz, CA, USA) (1:5000), monoclonal anti-Fpn 31A5, (Amgen) (1:10,000), polyclonal anti-Ftn (sc25617, Santa Cruz, CA, USA) (1:10,000), polyclonal anti-HCP (A0031, Dako, Santa Clara, CA, USA) (1:10,000), anti-hephaestin (sc-365365, Santa Cruz, CA, USA) (1:10,000), anti-DMT-1 (sc-166884, Santa Cruz, CA, USA) (1:10,000) and monoclonal anti-actin (sc1616, Santa Cruz, CA, USA) (1:10,000). After incubation with the appropriate secondary horseradish peroxidase-conjugated antibody, blots were developed with Enhanced ChemiLuminescence (ECL Prime) (GE Healthcare, UK). Protein levels were normalized on actin by densitometry analysis, performed with ImageJ.

### 2.8. Structures Preparations and Molecular Docking Simulations of the TfR1-Lfs Complexes

The X-ray structure of TfR1, at a resolution of 1.85 Å, was extracted from the PDB database (PDB ID: 6OKD) [18]. Three small missing loops in the structure were modelled using the Modeller 10.1 software (University of California, San Francisco, CA, USA), and the receptor was minimized in a box of TIP3P water molecules and 0.15 M of NaCl ions, using the ff19SB force field [19] and the AMBER16 software [20]. Representative structures of bLf and hLf were extracted from 50 ns MD simulations, performed using the ff19SB force field [18] and the AMBER16 software [20]. 

Blind protein–protein molecular docking simulations (i.e., no preferential sites were specified) between the TfR1 and the Lfs were performed using the CLUSPRO web-server (https://cluspro.bu.edu/home.php (accessed on 15 February 2022)) [21]. Hydrogen bonds and salt bridges were analyzed using the VMD hbond and salt-bridges modules, while non-polar contacts were identified using the contact_map routine of the mdtraj Python library [22]. The structure of the Tf-TfR1 complex, used for comparison to the hLf-TfR1 complex, was extracted from the PDB database (PDBID: 3S9L) [23]. Several Tf regions are missing in the crystallographic structure, including part of its C-lobe region.

### 2.9. Modelling of the SARS-CoV-2 Variant Structures

The SARS-CoV-2 Spike glycoprotein Alpha, Beta, Delta, and Omicron variant structures were modelled through Modeller 10.1, using as a reference the original Wuhan strain model used in our previous work [15,24]. We have taken advantage of the structure modelled in Romeo et al. [24] since the trimer Spike model, composed of three identical monomers, was already completed by modelling non-terminal missing loops. The mutations, insertions or deletions characterizing the different variants (Table 1) were introduced based on the data hosted on CoVariants.org, a web resource providing an overview of SARS-CoV-2 variants and mutations that are of interest coming from GISAID data.

### 2.10. Protein–Protein Docking Methods

As described, the Spike Alpha, Beta, Delta, and Omicron variant structures in prefusion conformation were modeled starting from that used in a previously published article [24]. The diferric form of bLf, refined at 2.8 Å resolution X-ray structure, was retrieved from the PDB database (PDB IDs: 1BLF) [25]. The protein–protein docking analyses between the Spike glycoproteins and the Lf structure were carried out using the Frodock docking algorithm [26], which combines the projection of the interaction terms into 3D grid-based potentials and the binding energy upon complex formation. A fast and exhaustive rotational docking search combined with a simple translational scanning was used to identify interaction-energy minima [27]. All the docking procedures were performed using Frodock’s (http://frodock.chaconlab.org/ (accessed on 10 April 2022)) webserver.

### 2.11. Molecular Dynamics

The tLeap module of AmberTools 21 (San Francisco, CA, USA) [28] was used to generate topology and coordinate files. Spikes and Lf were parametrized through the ff19SB force field [19] and inserted into a triclinic box of TIP3P water molecules, imposing a minimum distance of 12.0 Å from the box walls, while the solution was neutralized adding 0.15 mol/L of NaCl ions. To remove steric interactions, all structures underwent four minimization cycles of 500 steps of steepest descent followed by 1500 steps of conjugated gradient minimization. An initial restraint of 20.0 kcal mol-1 Å-2 was imposed on all protein atoms and subsequently reduced and removed in the final minimization cycle. Systems were progressively heated from 0 to 300 K in an NVT ensemble over a period of 5.0 ns using the Langevin thermostat, imposing an initial restraint of 0.5 kcal mol-1 Å-2 on all atoms, decreased every 500 ps to relax the system. The systems were simulated in an NPT ensemble for 2.0 ns, setting a pressure of 1.0 atm using the Langevin barostat and imposing the temperature at 300 K. Covalent bonds involving hydrogen atoms were constrained using the SHAKE algorithm [29]. 100 ns of production run were performed through the NAMD 2.13 MD code [30], using a 2.0 fs time step. The PME method was used to evaluate long-range interactions, while a cutoff of 9.0 Å was set for short-range interactions. Coordinates were saved every 1000 steps.

### 2.12. Trajectory Analysis

Hydrogen bond and salt bridges’ persistence were evaluated using the VMD hbond and salt-bridges modules coupled to in-house written codes, while distance analysis was carried out using the distance module of the GROMACS 2020.4 (Boston, MA, USA) analysis tools [31]. The non-polar contacts were identified using the contact_map and routines of the mdtraj Python library [22]. Generalized Born and surface area continuum solvation (MM/GBSA) analyses were performed over the last 30 ns of the trajectories, through the MMPBSA.py.MPI program as implemented in the AmberTools21 software (San Francisco, CA, USA) [20] on two nodes of the ENEA HPC cluster CRESCO6 [32]. Snapshots of the Spike-Lf complexes were generated using the UCSF Chimera program [33]. 

### 2.13. Statistical Analysis

For fusion experiments, Western blots and ELISA assays, statistically significant differences were assessed by one-way ANOVA and the post-hoc Tukey test. All statistical analyses were run using Prism v7 software (GraphPad Software, San Diego, CA, USA). Results were expressed as mean ± standard deviation (SD) of three independent experiments. A *p*-value ≤ 0.05 was considered statistically significant. 

## 3. Results

### 3.1. Lactoferrins Exert Neutralizing Activity against SARS-CoV-2 Spike Pseudovirus

The effect of different concentrations (1.25 and 6.25 μM, corresponding to 100 and 500 μg/mL) of bLf and hLf on Pseudovirus fusion with the cell membrane was initially tested on Vero E6 cells, an epithelial cell line largely used in SARS-CoV-2 studies, according to the experimental scheme described in Section 2.

The bovine protein exerted a strong inhibition of pseudoviral fusion in all experimental conditions tested, in particular when bLf was preincubated with Pseudovirus or added at treatment (Figure 1a,b). The human protein also induced a significant inhibition of pseudoviral fusion with Vero E6 cells at both concentrations tested, although its effect was weaker than that exerted by bLf (Figure 1c,d).

Lf concentrations were chosen following data in the literature. In particular, 1.25 μM is usually employed in anti-inflammatory studies [34,35,36], whereas higher concentrations are usually tested to disclose Lf antiviral properties [15,16,37,38,39]. Indeed, during infection and inflammation, Lf levels drastically increase in the biologic fluids, including blood, where Lf concentration is usually as low as 6–12 nM under healthy conditions, whereas it increases to 1.25–2.5 μM during systemic infections [40]. However, a higher dosage, achieved through Lf exogenous administration, could be requested to allow an efficient antiviral activity.

Preincubations had two complementary aims. On one hand, the order of addition of reagents can obviously give hints on the mechanism. On the other, preincubation of reagents allows the outcome of the measurement to be reasonably independent from kinetics of interaction. When Pseudovirus and Lf were preincubated, removal of unbound Lf was not attempted; therefore, we cannot in this case distinguish effects due to free Lf from those due to the eventual formation of a virus-Lf complex. As shown by the body of our results, this aspect turned out to be essentially irrelevant.

To prove that the effects of Lfs were also reproducible on a cell type extensively targeted by SARS-CoV-2, the experiments were carried out using the higher dose of Lf on the human bronchial epithelial 16HBE14o- cell line. Although with less efficacy when compared to Vero E6, both bLf and hLf were able to interfere with pseudoviral fusion in respiratory cells. Compared to hLf, bLf was more efficient when pre-incubated with the Pseudovirus than with the cells (Figure 2a,b). To corroborate our results also on a cell type primarily involved in iron homeostasis and to mimic the oral administration of these proteins, we applied the same experimental scheme to intestinal epithelial Caco-2 cells. As shown in Figure 2c,d, results comparable to Vero E6 and 16HBE14o- cells were obtained. Again, bLf (Figure 2c) proved to be more efficient than hLf (Figure 2d) in attenuating Pseudovirus fusion with cells. Similar results were obtained on differentiated macrophagic THP-1 cells (Figure 2e,f).

### 3.2. Lactoferrins Bind to SARS-CoV-2 Spike

To test whether bLf and hLf directly bind to SARS-CoV-2 Spike, an in vitro pull-down assay was set up. As a control of binding specificity, we used hTf, which belongs to the same family of Lactoferrins. We therefore prepared bLf-, hLf- and hTf-conjugated Sepharose 6B and the resins were incubated with either full-length stabilized trimeric Spike or with its S1 domain. As shown in Figure 3a, when SDS-eluted fractions of both bLf- and hLf-conjugated resins were probed with an anti-His Antibody (tag for Spike glycoprotein), an immuno-reactive band with molecular mass slightly lower than 250 kDa was present. According to the manufacturer’s datasheet, the band corresponds to the post-translationally modified monomeric form of the full-length viral glycoprotein. No reactive bands were recorded for both unconjugated and hTf-conjugated resins, demonstrating the specific binding between trimeric Spike and bLf/hLf. Of note, the S1 domain of Spike was not sufficient to bind Lfs, as shown by the absence of immunoreactive bands in the corresponding SDS-eluted fraction (Figure 3b). Therefore, we experimentally demonstrate, for the first time, that bLf is able to bind to Spike glycoprotein and that such interaction is dependent on its oligomerization state.

### 3.3. Bovine Lactoferrin Counteracts the Dysregulation of Iron Proteins Induced by SARS-CoV-2 Spike

To shed some light on a direct role of Spike on iron and inflammatory disorders and on the potential protective effect of Lf, the expression of the main iron-handling proteins and of interleukins involved in iron homeostasis has been evaluated in both enterocytes and macrophages challenged with purified Spike. As a matter of fact, purified SARS-CoV and SARS-CoV-2 Spike glycoproteins have been already proven to be potent inducers of IL-6 signaling [5,41], one the major regulator of systemic iron homeostasis. 

For these experiments we chose to selectively use bLf, which has proved to be more efficient than hLf in inhibiting Pseudoviral fusion with host cells (see above), has bioavailability and functions totally superimposable to those of hLf and, above all, has a definitely higher commercial availability, which makes it more convenient not only for in vitro, but also for in vivo studies, including clinical trials. 

Caco-2 and THP-1 cells were treated with 20 nM full length SARS-CoV-2 Spike in the absence or presence of 1.25 μM bLf, according to the experimental scheme described in Section 2. Figure 4 reports a representative Western blot (panel a) and the relative densitometries (panels b–f) on Caco-2 cells. Spike down-regulated Fpn, Heph and DMT-1 (Figure 4b,c,e), reaching statistical significance in the case of Heph and DMT-1. No effect on TfR1 and Ftn was observed (Figure 4d,f). BLf efficiently counteracted the Spike-induced dysregulation of iron proteins. For Fpn, Heph and DMT-1 the effect was invariably evident when a preincubated mixture of bLf and Spike was employed, suggesting that the two proteins likely interact. For Fpn and Heph, significant effects were recorded also for bLf pre-treated cells, whereas for Heph and DMT-1 the effect of bLf was significant also on cells treated with bLf 1 h after addition of Spike (Figure 4). This latter result suggests that bLf can reverse the effects of Spike even after they have been triggered. No detectable levels of IL-1β and IL-6, the main cytokines involved in iron disorders, were recorded both in basal and Spike/bLf-stimulated conditions (data not shown).

Data on macrophagic THP-1 cells are reported in Figure 5, with a representative Western blot in panel a. As observed with Caco-2 (shown in Figure 4), Spike induced a significant down-regulation of the iron exporter Fpn and, again, bLf easily counteracted the effect in all conditions tested (Figure 5b). We also measured the molecular partner of Fpn, namely the membrane-bound ferroxidase Cp, which was found to be positively affected by Spike treatment (Figure 5c). BLf was able, also in this case, to restore basal Cp levels. As for Caco-2 (shown in Figure 4), no Ftn modulation was detected upon Spike challenge (Figure 5e), while, at variance with the intestinal cells, a significant up-regulation of TfR1 (reversed by bLf) was observed in this case (Figure 5d). As expected, levels of IL-1β and IL-6 were easily detectable in macrophagic cells. As shown in Figure 5f, Spike induced a significant up-regulation of both interleukins and bLf counteracted the increase, its effect being significant when the bLf was preincubated with Spike and, for IL-6, even when added to cells before Spike.

### 3.4. TfR1 Contributes to SARS-CoV-2 Spike Pseudovirus Fusion to Cell Membrane

As already stated, it has been widely demonstrated that Lf blocks viral entry by competing with the virus structure and/or cell surface receptors. Moreover, SARS-CoV-2 has been reported to exploit multiple cell surface receptors for its entry, including TfR1 [42]. However, to date no data have been reported in our cellular models on the possible contribution of TfR1 in the entry of SARS-CoV-2. To explore this hypothesis further, we performed the pseudoviral neutralization assay in the presence either of a monoclonal antibody recognizing the ectodomains of human TfR1 or of a soluble form of TfR1. Bronchial and intestinal epithelial cells, as well as a macrophagic cell line, were used. Both anti-TfR1 antibody and soluble TfR markedly reduced pseudoviral fusion in all three cell lines (Figure 6). A significantly stronger effect of soluble TfR1 vs. anti-TfR1 antibody was observed in respiratory 16HBE14o- cells (Figure 6a).

### 3.5. Molecular Docking Simulations of TfR-1 in Complex with Lactoferrins

The involvement of TfR1 in Pseudovirus entry into host cells prompted us to evaluate the possibility that the blocking of Spike-mediated viral entry could also be linked to Lf competition with TfR1. The rationale was the knowledge of the high identity between Lf and Tf, the natural TfR1 interactor. On this basis, we performed molecular docking simulations between TfR1 and Lfs.

Main molecular docking binding pose obtained for the TfR1-hLf complex is reported in Figure 7a. In this binding pose, hLf localizes at the helical and protease-like domains of TfR1, at the interface of the two monomers, almost completely overlapping the Tf binding site on TfR1, as determined by X-ray crystallography (PDB ID: 3S9L) [23] (Figure 7b and Figure 8).

HLf interacts with 48 TfR1 residues, establishing six hydrogen bonds and 11 salt bridges (Table 2). In particular, 17 of these residues are also contacted by Tf in the crystallographic Tf-TfR1 complex (Table 2), confirming that both proteins contact equivalent regions on the TfR1 surface (Figure 8).

Main molecular docking binding pose obtained for the TfR1-bLf complex is reported in Figure 9. In this complex, bLf contacts the apical domain of TfR1 with its N-terminal lobe. A crystallographic structure retrieved from the PDB (PDB ID: 3KAS) [43] revealed that this TfR1 region is also the binding site of the trimeric GP1 surface glycoproteins of the MACV, JUNV, GTOV and SABV arenaviruses, responsible for hemorrhagic fevers in humans. The binding of GP1 surface glycoproteins to TfR1 allows virus internalization into endosomes [43]. The interactions established by bLf at this site, including 12 non-polar contacts and seven salt bridges, are reported in Table 2.

### 3.6. Computational Results on Bovine Lactoferrin and Spike Variants

The molecular docking simulations between bLf and the four Spike variants of interest (Alpha, Beta, Delta, and Omicron) indicate a preferential binding pose in which the bLf structure interacts with the RBD domain in the up conformation (Figure 10). For all the four docking simulations, the first three solutions obtained by the docking clustering procedure account for more than 60–70% of the total generated complexes, which are superimposable to the binding pose obtained in our previous work [15]. Using as a starting structure the first solutions obtained from docking experiments, we performed four 100 ns long classical MD simulations in order to verify the stability of the complexes, check for the presence of persistent interactions, and verify the ability of bLf to interact with all the Spike variants regardless of the number and position of the mutations. 

As shown in Figure 11, the distance between the centers of mass of the four Spike glycoproteins and bLf, calculated as a function of time, oscillates around the value of 4.5 nm, indicating a constantly close contact between the two molecules for all the simulation time. 

MM/GBSA analyses confirmed the high affinity of the bLf for the Spike glycoprotein (Table 3), showing an interaction energy of −36.2, −69.1, −46.4 and −45.8 kcal/mol for the Alpha, Beta, Delta, and Omicron Spike variants, respectively. Interestingly, MM/GBSA results underlined that the energy term mainly contributing to the binding energy switches from the Van der Waals term for the Alpha variant (as observed for the Wuhan isolate) [15] to the polar solvation term for the Omicron. This suggests that, although the recognition occurs with similar orientations of the interacting partners, the detected interactions defining the complexes significantly differ for the four studied variants. 

A detailed analysis of the interaction networks, reported in Table 4, revealed an increase in high-persistence hydrogen bonds and salt bridges between Spike and bLf, going from the Alpha variant to Omicron. For the Alpha variant, we observed 94 different interactions, which persist for more than 40% of the simulation time, consisting of 3 salt bridges, 4 hydrogen bonds and 87 residue pairs involved in non-polar contacts (Table 4 and Table 5, Alpha column). For the Beta variant, the number of high-persistence interactions increases to 113, with 4 salt bridges, 4 hydrogen bonds and 105 residue pairs involved in non-polar contacts (Table 4 and Table 5, Beta column). As expected from the MM/GBSA results, we observed an increase in polar and charged interactions for the Delta and Omicron variants, with five salt bridges and six and seven hydrogen bonds, respectively (Table 4, Delta and Omicron columns). On the other hand, there is a reduction in non-polar contacts, with 100 and 70 residue pairs involved in these interactions (Table 5, Delta and Omicron columns). Remarkably, in the case of Omicron, four out of five reported salt bridges involve variant-specific mutations. 

These results allow us to hypothesize that bLf should retain its ability to bind the surface of the Spike glycoprotein, independently of the mutations observed for the variants of concern that have emerged so far.

## 4. Discussion

The severity of CoV infections is mainly regulated by and dependent on the Spike glycoprotein, which, along with cell tropism and infectivity, regulates viral spread and host responses. Although the receptor-binding domains of Spike from SARS-CoV-2 and SARS-CoV share ca. 75% amino acid identity, the two viruses show significant differences in their ability to infect and transmit in humans [44]. Interestingly, recent papers have highlighted the possible role of Spike in contributing to the higher virulence of SARS-CoV-2 [44]. Indeed, Spike is emerging as the main virulence factor of SARS-CoV-2, able to induce host immunopathogenesis, which is, in turn, the critical regulator of virus infection and disease outcomes [44]. For this reason, all efforts in the last two years have focused on discovering substances capable of interacting with Spike and, in turn, inhibiting SARS-CoV-2 infection.

In this respect, in silico results reported in the paper by Campione et al. [15] had pointed to Lf as an ideal candidate for counteracting SARS-CoV-2 infection due to its putative ability to bind to the C-terminal domain of Spike. Here, we experimentally demonstrate for the first time that Lf and SARS-CoV-2 Spike actually interact. From a functional point of view, we validated the hypothesis by investigating the neutralizing activity of human and bovine Lf against a Pseudovirus decorated with the SARS-CoV-2 Spike protein, in three epithelial and a macrophagic cell lines. The results clearly show that Pseudovirus fusion with cells is invariably inhibited by Lfs, with minor variations in terms of concentration dependence, Lf source, experimental protocol, and cell line. It is interesting to note that bLf exerts a more potent inhibition compared to hLf. The highest decrease in Pseudoviral fusion was observed when bLf and Pseudovirus were added together, with or without pre-incubation. This is a good indication that bLf may physically interact with Spike and that this can be one of the molecular mechanisms at the basis of the inhibitory effect exerted by Lfs against SARS-CoV-2. In other terms, bLf hinders Spike-mediated virus entry by competitive inhibition of Spike-mediated virus binding to host receptors, with an efficacy likely depending on cell-specific expression of different plasma membrane receptors in different cell lines which modulate SARS-CoV-2 entry rate. 

As already reported, purified Spike glycoproteins from SARS-CoV and SARS-CoV-2 have been shown to be potent inducers of pro-inflammatory response in macrophage [41] and epithelial [5] cells. In particular, purified Spike from SARS-CoV was found to induce the up-regulation of pro-inflammatory cytokines, namely IL-6 and IL-8, via the activation of the NF-κB pathway in both human peripheral blood monocyte macrophages and THP-1 cells [41]. Similarly, Spike from SARS-CoV-2 was demonstrated to promote NF-κB and AP-1/c-Fos pathways via MAPK activation in epithelial cells, thus inducing the downstream release of IL-6 [5]. The knowledge that up-regulation of NF-κB can lead to overexpression of TfR1 [45] and that Fpn usually decreases when IL-6 increases can reconcile our data.

In this context, the role of iron, a transition metal involved in many fundamental biological processes, including DNA/RNA synthesis and ATP generation, must be considered. As a matter of fact, higher iron availability, strictly associated with inflammatory disorders, has been shown to promote viral spread, which requires active cell metabolism, as demonstrated for the human immunodeficiency virus (HIV) [46], where it is involved in several key steps of virus replication, from the reverse transcription process to the iron-dependent production of dNTPs [47]. Moreover, iron is implicated in the activation of NF-κB signaling by generation of reactive oxygen species (ROS) [48]. Recently, it has been also reported that SARS-CoV-2 replication is dependent on host cell iron-related enzymes, some of which are involved in transcription, viral mRNA translation and viral assembly [49]. 

Here, for the first time, we demonstrate that SARS-CoV-2 Spike glycoprotein can induce dysregulation of some of the main iron-handling proteins, both in enterocytes and macrophages. In particular, a significant down-regulation of Fpn was found in both models tested, thus suggesting a possible induction of intracellular iron retention. This hypothesis was further corroborated by the down-regulation of DMT-1 and Heph in the enterocytes and by the up-regulation of TfR1 in the macrophagic model. Overall, the observed changes agree with previous studies on iron proteins and inflammation, particularly with the acute phase response [6]. Indeed, the decrease in Fpn expression upon inflammatory challenge has been widely reported in several in vitro models [50,51] and confirmed in animal studies [52,53]. Such an effect is usually linked to the induction of pro-inflammatory cytokines, in particular IL-1β and IL-6, in partnership or not with the hepcidin pathway [54]. Notably, Spike treatment has induced opposite effects on the two Fpn functional partners, namely Cp and Heph. While a down-regulation of Heph was observed in enterocytes, Cp turned out to be up-regulated by Spike challenge in macrophages. To understand this apparent discrepancy, we should recall that, besides a ferroxidase ability in common with Heph, Cp has been shown to be endowed with multiple functions, ranging from copper transport to biological amine oxidation, to antioxidant activity exerted through several different mechanisms [10]. Not surprisingly, Cp and Heph are differently regulated at the translational level [55]. Cp is an inflammatory, acute-phase plasma protein produced by hepatocytes and monocyte/macrophages, induced by inflammation or iron loading [56]. The cell type-specific regulation of Cp expression has been demonstrated in myeloid lineages, showing Cp synthesis to be successfully induced by TNF-α in alveolar macrophages [57] and in monocytic cell lines by IFN-γ [58]. Moreover, Persichini et al. [59] reported the up-regulation of Cp, both in the secreted and GPI-linked forms, upon treatment with IL-1β in a glial cellular model [59]. Our results lie within this framework, with the Spike protein able to induce Cp expression in accordance with its ability to up-regulate pro-inflammatory response in THP-1 cells. On the other hand, Heph (and DMT-1) down-regulation in Caco-2 cells is consistent with a pro-inflammatory challenge, as already reported in some studies [60,61]. Regarding TfR1, its significant up-regulation in THP-1 cells after Spike treatment is in line with other studies reporting increased expression during the acute phase response [45,62]. Despite its role in iron uptake, TfR1 is also hijacked by numerous viruses to enter the cell [11], and SARS-CoV-2 does not seem to be an exception [42]. Therefore, TfR1 up-regulation can both act as a SARS-CoV-2 gate for its cell entry and favor viral metabolism and replication through iron intake. In this respect, data in the present study corroborate the results obtained by Tang et al. [42], confirming the contribution of TfR1 in Spike-mediated cell fusion in bronchial and intestinal epithelial as well as in macrophagic cell lines. Intriguingly, despite being cellular types involved in iron retention, intracellular Ftn did not significantly increase in either enterocytic nor macrophagic cells. Hepatocytes, macrophages and Kupffer cells have been shown to secrete Ftn [63], a process enhanced by iron and pro-inflammatory cytokines [64]. Interestingly, Tran et al. [64] demonstrated, in a murine model of acute phase response, that the administration of IL-1β or TNF-α doubled the amounts of secreted Ftn, while it did not influence intracellular Ftn levels [64]. New recent evidence on COVID-19 patients shows that serum Ftn levels are increased as the disease worsens, providing a potential indication of the mortality risk [65,66]. However, despite the robust association with mortality, whether hyper-ferritinemia is a mere systemic marker of disease progression or a key modulator in disease pathogenesis has not been yet clarified.

In this scenario, we demonstrated that bLf counteracts both the up-expression of IL-6 and the dysregulation of the main iron-handling proteins in different conditions. Beyond the blockade of Spike-induced pathogenesis through direct binding, here demonstrated for the first time in vitro, and competition with cell surface receptors when pre-incubated with both cell monolayer or the viral glycoprotein, bLf can rebalance these iron and inflammatory disorders even when intervening after the Spike challenge.

In recent years, our group has demonstrated the efficacy of bLf in reverting iron dysregulation in different inflamed/infected in vitro [51,67] and in vivo [68] models as well as in clinical trials [54]. Such an effect on iron homeostasis can be related to the ability of Lf to chelate free iron and downregulate pro-inflammatory cytokines, such as IL-1β and IL-6, thus boosting anti-oxidant and anti-inflammatory host response to viral infection. 

Beside the involvement of Spike in the dysregulation of iron proteins in both enterocytes and macrophages and the protective action of bLf on such effects, here we present strong evidence at a mechanistic level that both hLf and bLf can interact with the trimeric form of a full-length SARS-CoV-2 Spike glycoprotein. To date, the Lf efficacy in inhibiting SARS-CoV-2 entry was mostly associated with its ability to interact with the host cell surface molecules, of which heparan sulfates were demonstrated to be the most involved [14]. Through this study, we expanded our recently published in silico model [15] and validated it through an in vitro pull-down assay, allowing us to affirm that Lfs can consistently block SARS-Co-V-2 entry also through its direct binding to Spike. The possibility exists, of course, that Lf may bind other viral proteins encoded by SARS-CoV-2 in addition to Spike protein.

Furthermore, to atomistically understand the molecular mechanisms through which Lfs can block viral entry, we performed molecular docking simulations with TfR1, a secondary receptor for SARS-CoV-2 entry. In this regard, we unequivocally proved the involvement of TfR1 in viral entry in our experimental models. The bioinformatic data suggest that the TfR1 surface represents a promising binding site for both the human and bovine Lfs. Remarkably, the binding pose obtained for the human form strikingly resembles that adopted by Tf, the natural binder of TfR1, both in orientation and interaction pattern. BLf does not achieve the same binding pose but contacts a completely different region, the apical domain of TfR1, known to be the binding site of different “New World” arenaviruses. The binding of these viruses to the TfR1 apical domain allows their internalization into the host cell. However, despite these promising docking results and the high sequence and structural similarity between Tf and Lfs, it should be noted that experimental evidence suggests very low levels of binding between bLf and TfR1 [69].

Finally, the results obtained from molecular docking between bLf and Spike variants strongly suggest that the ability of bLf to interact with Spike is not influenced by single point mutations that occur in the more widespread genetic variants. This finding further promotes the potential use of bLf in the early stages of SARS-CoV-2 infection.

In conclusion, our data corroborate the results of our preliminary clinical trials [70,71] where, with the caveat of the limited number of Lf-treated patients, it was observed that a prompt bLf treatment decreases (i) the time to SARS-CoV-2 RNA negativization (14–15 versus 24–28 days) [70,71]; (ii) the clinical symptoms’ recovery [71]; (iii) serum IL-6, Ftn and D-dimer levels [71]. In addition, a very interesting link between symptom reduction and age was observed: the protective effect of Lf in reducing the time to symptom resolution is related to advancing age [70]. 

Further studies need to be carried out to ascertain the in vivo efficacy of Lfs in SARS-CoV-2 infection and sequelae. However, our findings give hope for its use as a readily available adjuvant to standard therapies in the treatment of COVID-19. 

## 5. Conclusions

Our results clearly demonstrate the actual binding between Lf and Spike and, hence, that Lf is able to interfere with Spike-mediated pseudoviral entry and Spike-induced iron dysregulation, thus giving hope for the use of bovine lactoferrin, already available as a nutraceutical, as an adjuvant to standard therapies in COVID-19. 

## Figures and Tables

**Figure 1 pharmaceutics-14-02111-f001:**
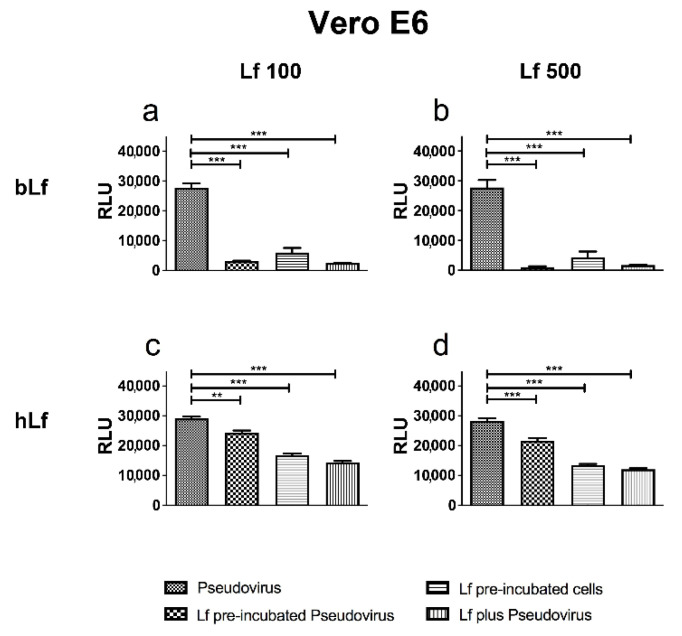
Luminescence of Pseudovirus observed in Vero E6 cells infected at multiplicity of infection (MOI) of 10 in the presence or absence of 1.25 (**a**,**c**) or 6.25 μM (**b**,**d**) of bovine lactoferrin (bLf) (**a**,**b**) or human lactoferrin (hLf) (**c**,**d**). See text for details. Data represent the mean values of three independent experiments. Error bars: standard error of the mean. Statistical significance is indicated as follows: **: *p* < 0.01; ***: *p* < 0.001 (one-way ANOVA with post-hoc Tukey test). RLU = Relative Luminescence Units.

**Figure 2 pharmaceutics-14-02111-f002:**
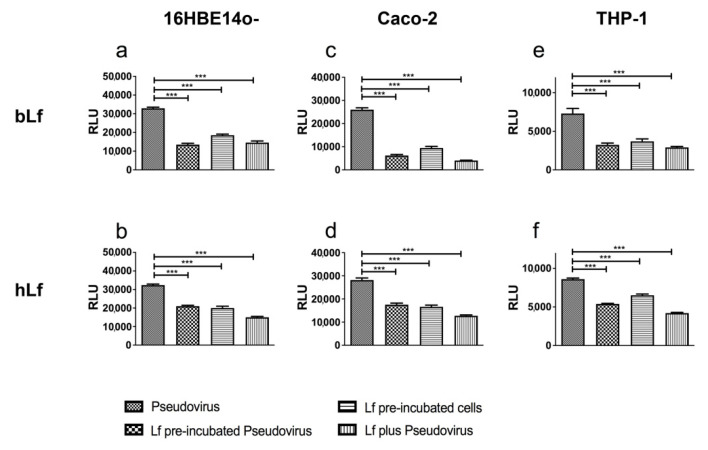
Luminescence of Pseudovirus observed in 16HBE14o- (**a**,**b**), Caco-2 (**c**,**d**) and THP-1 (**e**,**f**) cells infected at multiplicity of infection (MOI) of 10 in the presence or absence of 6.25 μM of bovine lactoferrin (bLf) (**a**,**c**,**e**) or human lactoferrin (hLf) (**b**,**d**,**f**). See text for details. Data represent the mean values of three independent experiments. Error bars: standard error of the mean. Statistical significance is indicated as follows: ***: *p* < 0.001 (one-way ANOVA with post-hoc Tukey test). RLU = Relative Luminescence Units.

**Figure 3 pharmaceutics-14-02111-f003:**
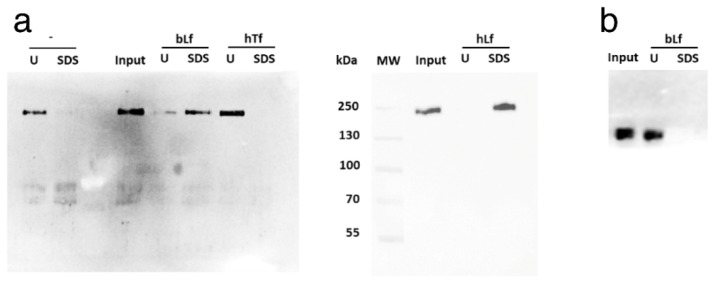
Sepharose 6B pull down of full-length stabilized trimeric (**a**) and S1 domain (**b**) SARS-CoV-2 Spike. Unconjugated (-), bovine Lactoferrin (bLf)-, human Lactoferrin (hLf)- and human Transferrin (hTf)-conjugated Sepharose 6B resins were employed. Input, unbound (U) and SDS eluted fractions were analysed through Western blot.

**Figure 4 pharmaceutics-14-02111-f004:**
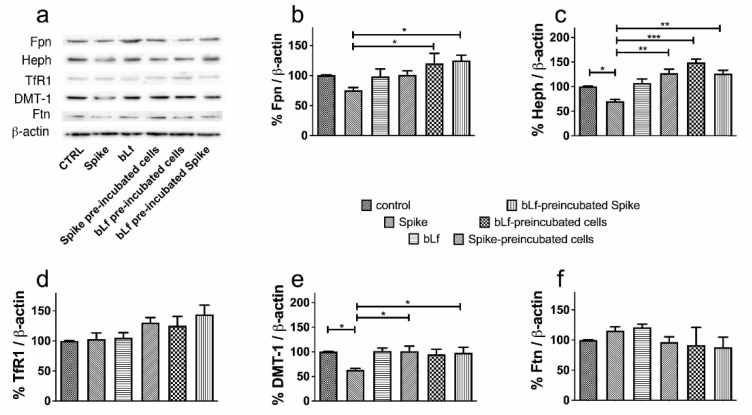
Protective effect of bovine lactoferrin (bLf) against iron and inflammatory disorders induced by SARS-CoV-2 Spike glycoprotein on Caco-2 cells. Western blot (panel (**a**)) and densitometry analysis of ferroportin (Fpn) (**b**), hephaestin (Heph) (**c**), transferrin receptor 1 (TfR1) (**d**), DMT-1 (**e**) and ferritin (Ftn) (**f**) levels in Caco-2 cells untreated or treated with 20 nM Spike glycoprotein in the absence or presence of 1.25 μM bLf. See text for details. Error bars: standard error of the mean. Statistical significance is indicated as follows: *: *p* < 0.05; **: *p* < 0.01; ***: *p* < 0.001 (one-way ANOVA with post-hoc Tukey test).

**Figure 5 pharmaceutics-14-02111-f005:**
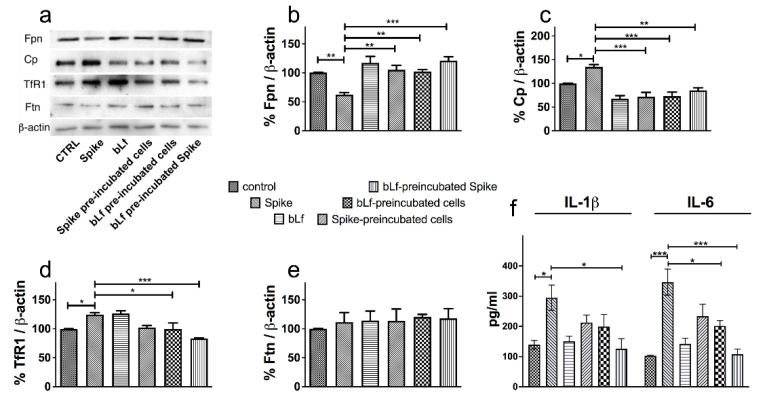
Protective effect of bovine lactoferrin (bLf) against iron and inflammatory disorders induced by SARS-CoV-2 Spike glycoprotein on THP-1 cells. Western blot (**a**) and densitometry analysis of ferroportin (Fpn) (**b**), membrane-bound ceruloplasmin (Cp) (**c**), transferrin receptor 1 (TfR1) (**d**) and ferritin (Ftn) (**e**) levels and ELISA quantitation of IL-1β and IL-6 production (**f**) in THP-1 cells untreated or treated with 20 nM Spike glycoprotein in the absence or presence of 1.25 μM bLf. See text for details. Error bars: standard error of the mean. Statistical significance is indicated as follows: *: *p* < 0.05; **: *p* < 0.01; ***: *p* < 0.001 (one-way ANOVA with post-hoc Tukey test).

**Figure 6 pharmaceutics-14-02111-f006:**
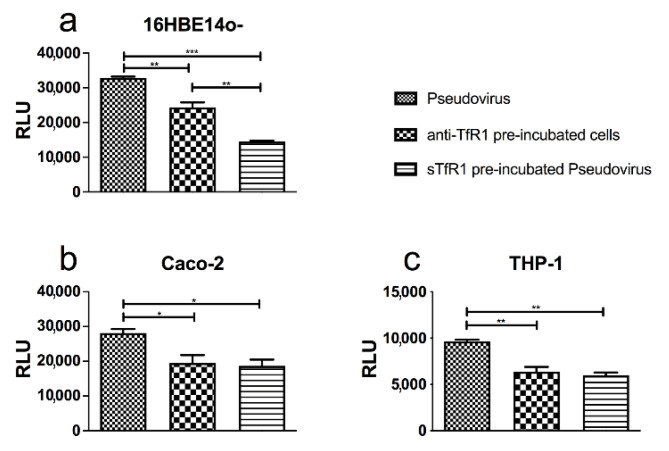
Luminescence of Pseudovirus observed in in 16HBE14o- (**a**), Caco-2 (**b**) and THP-1 (**c**) cells infected at multiplicity of infection (MOI) of 10 in the presence or absence of 200 nM monoclonal antibody recognizing the ectodomains of human transferrin receptor 1 (TfR1) (anti-TfR1) or 200 nM soluble human TfR1 (sTfR1). See text for details. Data represent the mean values of three independent experiments. Error bars: standard error of the mean. Statistical significance is indicated as follows: *: *p* < 0.05; **: *p* < 0.01; ***: *p* < 0.001 (one-way ANOVA with post-hoc Tukey test). RLU = Relative Luminescence Units.

**Figure 7 pharmaceutics-14-02111-f007:**
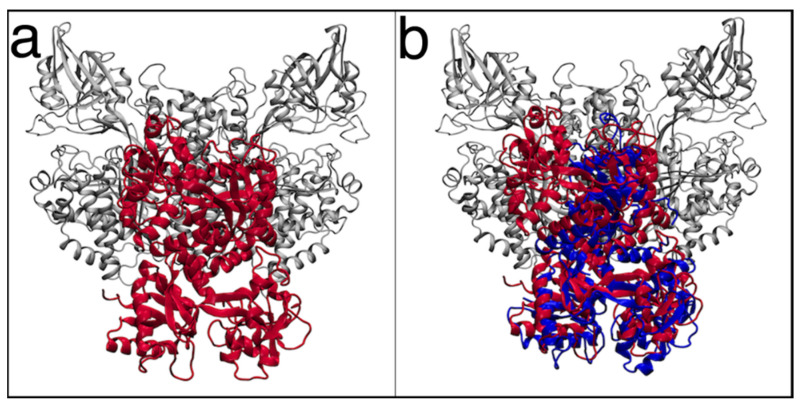
(**a**) Complex between human lactoferrin (hLf) (in red) and transferrin receptor 1 (TfR1) (in grey) obtained through molecular docking simulations. (**b**) Superposition of the hLf-TfR1 docking pose with the Tf-TfR1 crystallographic structure (PDB ID: 3S9L).

**Figure 8 pharmaceutics-14-02111-f008:**
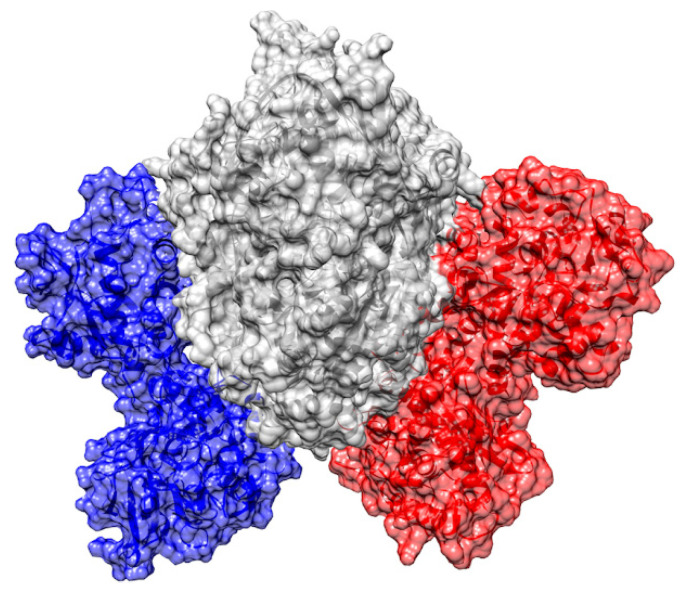
Comparison between the transferrin and transferrin receptor 1 (Tf-TfR1) crystallographic structures (PDB ID: 3S9L) and the obtained human lactoferrin (hLf)-TfR1 docking pose. Both TfR1 monomers are represented in grey; Tf is represented by a blue transparent surface bound to TfR1 monomer A, while hLf is in red bound on TfR1 monomer B. Both TfR1 monomers are equivalent for sequence, structure and interactions, Tf and hLf can therefore occupy the same location.

**Figure 9 pharmaceutics-14-02111-f009:**
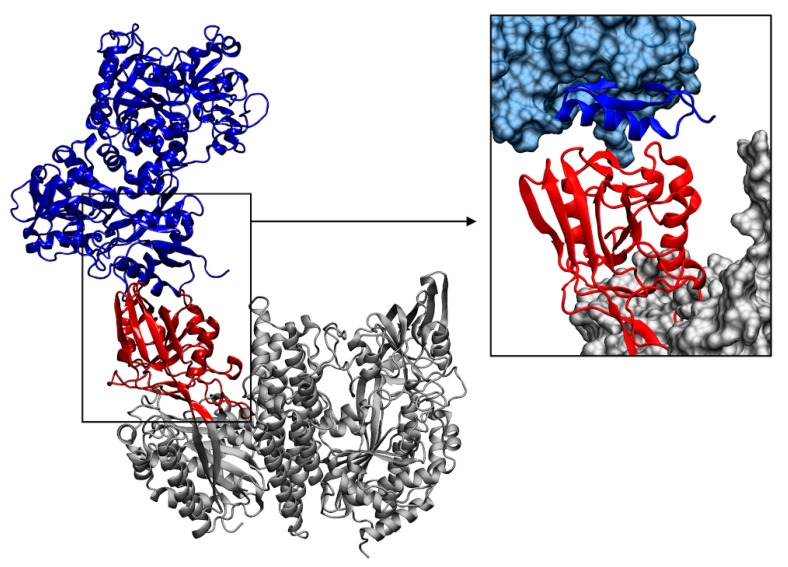
Complex obtained between bovine lactoferrin (bLf) (in blue) and human transferrin receptor 1 (TfR1) (in grey). The apical domain of TfR1, binding site of bLf, is highlighted in red. A closer representation of the interaction site is shown in the right image, where the proteins are represented as a solid surface, except for the interacting regions that are shown as cartoons.

**Figure 10 pharmaceutics-14-02111-f010:**
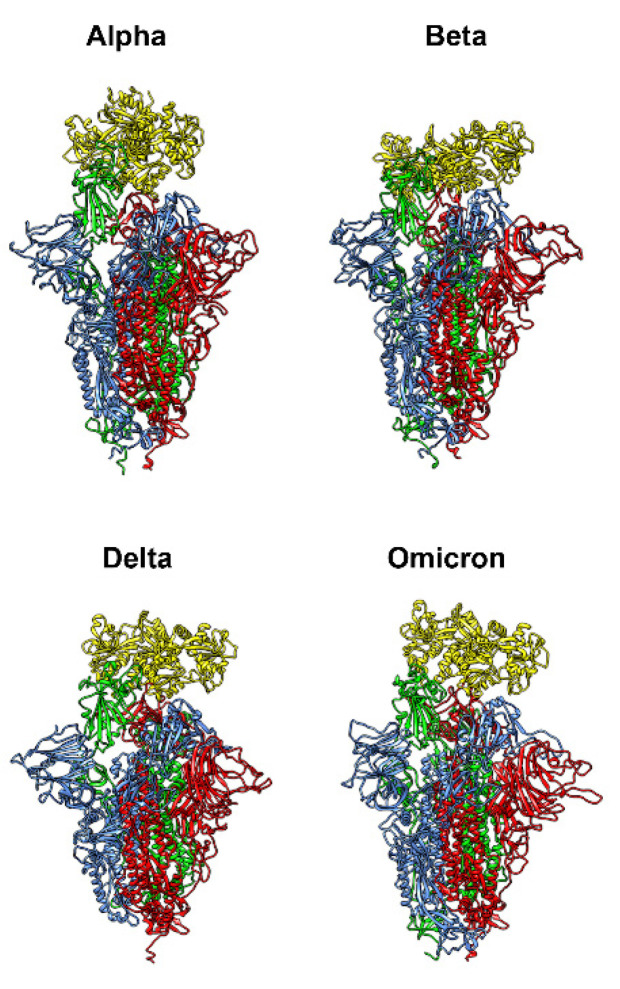
Complex between bovine lactoferrin (bLf) and the four Spike variants of interest obtained through molecular docking simulations. bLf is represented with yellow ribbons, while the three different chains composing the Spike glycoprotein are represented by red, blue, and green ribbons.

**Figure 11 pharmaceutics-14-02111-f011:**
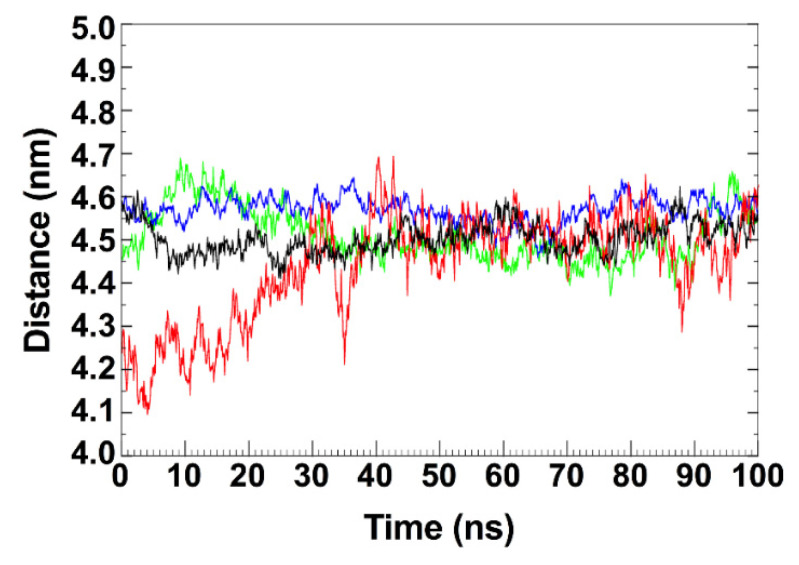
Time-dependent analysis of the distance evaluated between the centers of mass of the bovine lactoferrin and the receptor binding domain in the up conformation of the four Spike variants (black: Alpha, red: Beta, green: Delta and blue: Omicron).

**Table 1 pharmaceutics-14-02111-t001:** The mutations, insertions or deletions characterizing the different variants simulated in this work.

Variant of Concern	Defining Mutations
B.1.1.7 (Alpha)	ΔH69-V70, ΔY144, N501Y, A570D, D614G, P681H, T716I, S982A, D1118H
B.1.351 (Beta)	D80A, D215G, ΔL241-L242-A243, K417N, E484K, N501Y, D614G, A701V
B.1.617.2 (Delta)	T19R, ΔE156-F157, R158G, L452R, T478K, D614G, P681R, D950N
B.1.1.529 (Omicron)	A67V, ΔH69-V70, T95I, ΔG142-V143-Y144, Y145D, ΔN211, L212I, G339D, S371L, S373P, S375F, K417N, N440K, G446S, S477N, T478K, E484A, Q493R, G496S, Q498R, N501Y, Y505H, T547K, D614G, H655Y, N679K, P681H, N764K, D796Y, N856K, Q954H, N969K, L981F

**Table 2 pharmaceutics-14-02111-t002:** Hydrogen bonds, salt bridges and non-polar interactions established between transferrin receptor 1 (TfR1) (monomer A and B) and human lactoferrin (hLf) or bovine lactoferrin (bLf). TfR1 residues highlighted in grey are also contacted by Tf in the Tf-TfR1 crystallographic structure (PDB ID: 3S9L).

TfR1-hLf	TfR1-bLf
**Hydrogen bonds**	**Hydrogen bonds**
K385.B–Q512	/
R646.A–C371	/
S654.A–E388	/
R121.A–Q165	/
G661.A–Y65	/
Y123.A–Q165	/
**Salt bridges**	**Salt bridges**
D755.A–K73	D245.B–K27
K508.A–E335	D352.B–R21
K508.A–E336	D356.B–K28
E623.A -E366	K205.B–E176
R183.B–E514	R208.B–E178
K385.B–E514	E369.B–R186
E606.A–R332	E369.B–R38
E612.A–K333	/
R629.A–E637	/
K633.A–E637	/
E664.A–R120	/
**Non-polar contacts**	**Non-polar contacts**
S120.A: P167	D245: K28
Y123.A: P144, E146, A147, F166	Y247: R20, Q23, W24, K27
W124.A: T139, E143	T248: R20, W24
D125.A: E143, R151	E350: Q13, W16, F17, R20
K508.A: T139, S334	G351: F17, W24
Q511.A: E336	D352: R25, S285
K600.A: P142	C353: R25
N608.A: P142	P354: R25, K28
L619.A: N359, S362, G363, G367, T370	S355: R25
R623.A: Q360, G363, L364	M365: W24
D626.A: Q360	V366: R20, W24
R629.A: S636	E369: W16
Q640.A: E352, E353, R356	/
Y643.A: L355, R356, N359	/
R646.A: L355, N359	/
G647.A: L355	/
F650.A: V346, T370, C371, S372	/
R651.A: S373	/
T658.A: E388	/
F660.A: R332	/
G661.A: I328	/
D662.A: Y65, L69	/
A663.A: L69	/
E664.A: L69	/
K665.A: R332	/
V670.A: A70	/
E163.B: Q512, G513, E514	/
K177.B: N52, N261	/
Q185.B: E514	/
F187.B: Q512	/
K394.B: P71, Y72	/

**Table 3 pharmaceutics-14-02111-t003:** Results of the MM/GBSA analyses performed over the last 30 ns of the Spike-bovine lactoferrin complex simulation for the four variants of concern.

MM/GBSA Results					
Variant	VdW (kcal/mol)	Electrostatic (kcal/mol)	Nonpolar Solvation (kcal/mol)	Polar Solvation (kcal/mol)	ΔG_binding_ (kcal/mol)
**B.1.1.7**	−181.5 ± 11.1	−22.9 ± 47.6	−22.6 ± 1.8	190.7 ± 49.5	−36.2 ± 8.8
**B.1.351**	−175.3 ± 12.4	280.1 ± 56.0	−22.5 ± 1.8	−151.4 ± 54.8	−69.1 ± 13.5
**B.1.617.2**	−164.9 ± 9.2	473.13 ± 56.5	−22.1 ± 1.2	−332.4 ± 53.1	−46.4 ± 8.3
**B.1.1.529**	−156.7 ± 10.1	836.9 ± 51.6	−21.6 ± 1.1	−758.8 ± 47.7	−45.8 ± 11.0

**Table 4 pharmaceutics-14-02111-t004:** Salt bridges and hydrogen bond interactions established between the bovine lactoferrin and the surface of the Spike glycoprotein (chain A, B or C), calculated for the four variants of concern. Only interactions identified for more than 40% of simulation time are shown. Residues highlighted in grey corresponds to variant-specific mutations.

B.1.1.7 (Alpha)	B.1.351 (Beta)	B.1.617.2 (Delta)	B.1.1.529 (Omicron)
**Salt bridges**			
D443–K458A	K358–D398A	K358–D405A	D646–K440B
D126–K444A	E352–K444C	K642–D467A	E355–R498C
E574–K444B	E407–K444C	E355–R409A	E356–R498C
	D646–K444B	D646–K444B	E654–R498B
		E355–K378A	K358–D405A
**Hydrogen bonds**			
N349–Y501A	E356- N449C	Q386–N439C	K358–D405A
Q628–E406A	S160–F490C	N387–T500C	E356–R498C
H439–Y439A	S381–N440C	D646–K444B	E355–R498C
Q386–N465C	Q378–N440C	Q386–Q506C	E356–S496C
		C390–N440C	D646–K440B
		E355–R408A	R363–F497C
			Q386–Q506C

**Table 5 pharmaceutics-14-02111-t005:** Non-polar interactions established between the bovine lactoferrin and the Spike glycoprotein surface (chain A, B or C), calculated for the four variants of concern. Only interactions identified for more than 40% of simulation time are shown.

B.1.1.7 (Alpha)	B.1.351 (Beta)	B.1.617.2 (Delta)	B.1.1.529 (Omicron)
**Non-Polar Contacts**			
K627: D402A, R405A, Q411A, T412A, G413A	C377: W436C, N437C, S438C, N439C, N440C	G641: F462A, R464A	T576: F494B
L630: Q411A	S381: W436C, S438C, Y505C	K642: N437B	C644: N436B
L631: P409A, G410A, Q411A, P497C	W380: S438C, N439C	C644: S441B, P497B	P645: N436B
H632: P497C	G385: S438C, G496C, F497C, Q498C, P499C, T500C, Y505C	P645: S436B, N437B, D440B, S441B, F495B, Q496B, P497B, T498B	S653: F494B, R495B, P496B
Q634: Q411A	K376: N439C	D646: S436B, N437B, P497B, T498B, Y503B	L151: G499A
A635: P497C, T498C	N387: N439C	K647: P497B	R152: R495A, P496A, T497A, Y498A, G499A
L636: T498C	V388: N439C, K444C, G496C	F648: P497B	P153: R495A, P496A, T497A
K642: T498C	T389: N439C, K444C	F651: F495B, Q496B, P497B	L155: R495A
P645: G499B	E373: N440C	S653: F495B, Q496B	S156: G499A
D646: T498C, G500C	E355: S443C, K444C, G446C, G447C, Y495C, G496C	L125: Q496A	W157: H502A
S653: P496B	T362: S443C, K444C, V445C, G446C, G447C, G496C, F497C, Q498C	L151: G500A	E159: C477C
A668: D417A	A359: K444C, V445C, G446C	R152: N499A, G500A, G502A, Y503A	S160: P476C, C477C, N484C
Q124: Q495A	V364: K444C, V445C	P153: Q496A, G502A, Y503A	T353: G499A, V500A
L125: S440A	C390: K444C	L155: W434A	A354: G401A, F494C
M148: P496A, T497A	G406: K444C, V445C	W157: C486C	E355: Y492C, F494C
G149: Q495A, P496A, T497A, G501A	L347: V445C	E159: C486C, L490C	V357: I399A, G401A, G499A, V500A, G501A
R152: G501A, Y502A	R351: V445C, G446C, G447C, N448C	S160: S369A, N485C, C486C	K358: F494C, R495C
P153: S435A, Q495A, G501A	E352: N448C	L251: P497A	Y361: R495C
C250: F494A, Q495A, P496A	A184: G476C	N252: Q496A, P497A	T362: S493C, F494C, R495C
L251: F494A, Q495A	S160: P479C, C480C, F486C, N487C, C488C, Y489C, S371A	L347: V443C	Q374: N436C
N252: Q495A	E159: C488C, Y489C, L492C	R351: G444C, N446C	C377: S435C, N436C, D439C
S341: T497A	K358: G496C, Q498C	E352: N446C	Q378: G499B
L344: T497A	S384: F497C, Q498C, P499C, T500C	T353: G500A	W380: S493C
T345: T497A	Y361: Q498C	A354: V405A	S381: Y498C
K348: T497A, Y498A	Q386: Q498C, P499C	E355: V405A, N446C, G494C	S384: F494C, R495C, P496C, T497C, Y498C
E356: G499A	Q383: P499C	V357: G402A, V405A	G385: S493C, F494C, R495C, P496C, T497C, H502C
V357: G499A	W157: S371A	K358: V405A, Q496C	Q386: R495C, P496C
K358: K442C, G494C	A354: F374A, V407A	Y361: P497C	N387: S493C
R360: T497A, Y498A, G499A	V357: V407A	T362: G445C, G494C, Q496C	V388: S493C
Y361: V404A, G494C, Q496C	P645: W437B, S439B, N440B, Y506B	E373: N437C	T389: D439C, S440C, K441C
T362: K442C, G494C	C644: S439B, N440B, S444B, Q499B	K376: N437C	C390: K441C
C377: N437C	D646: S439B, T501B, Y506B	C377: N435C, S436C, N437C, N438C	A391: K441C
S381: S436C, N437C, D440C, Y499C, Y503C	S575: F498B	Q378: P497B, T498B, N499B	V401: K441C
Q382: P497C, T498C, Y499C	S653: F498B, T501B	W380: S436C, N437C	G406: K441C
Q383: Q496C, P497C, T498C, Y499C	F648: Q499B, P500B	S381: G500B, W434C, S436C, Y503C	S575: V442B, F494B
S384: G494C, F495C, Q496C, P497C, T498C, Y499C	F651: Q499B, P500B	Q382: G500B, Y503B	
G385: S436C, D440C, K442C, G494C, Q496C	K375: P500B	S384: Q496C, P497C, T498C, N499C	
Q386: Q496C, P497C	Q378: P500B, T501B, Y502B, G503B	G385: W434C, S436C, G494C, F495C, Q496C, P497C, T498C, Y503C	
S437: E468A, I469A	K647: P500B	Q386: Q496C, P497C, T498C	
K438: E468A, I469A	K652: P500B	V388: N437C, G494C	
	Q382: G503B, Y506B	T389: N437C	
		C390: K442C	
		V401: K442C	
		G406: K442C, V443C	

## Data Availability

Original data are available on request. Please contact the corresponding author.

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
