# Peer review of "Lactoferrin Binding to SARS-CoV-2 Spike Glycoprotein Blocks Pseudoviral Entry and Relieves Iron Protein Dysregulation in Several In Vitro Models"

_pharmaceutics, 2022, doi:10.3390/pharmaceutics14102111_

Round 1

Reviewer 1 Report

This is a very interesting work of a current health problem worldwide, the infection with SARS-CoV2. The authors found that lactoferrin is able to bind and neutralize the spike protein from the virus, in a model of pseudovirus.

In general, the manuscript is well written and the work carefully explained. I suggest some recommendations in order to have an improved manuscript:

1. Use only micromolar concentrations of all reagents, mainly Lfs.

2. In Results do not repeat the conditions of the experiments, which were explained in Methods and are detailed in the figure legends.

 3. Add a squeme that includes the results and conclusions obtained, which could be more understandable and descriptive.

Reviewer 2 Report

The authors investigated the inhibitory effects of bovine and human Lf on SARS-CoV-2 infection in cell culture. “It has been widely demonstrated that Lf blocks viral entry by competing with the virus structure and/or cell surface receptors (lines 453-454).”  This study demonstrated that Lf interacts with the spike protein and took molecular docking approaches for verification and structural analyses.

Major

1. The concentrations of Lf used in this study are not well justified. References cited (12, 13, 15, 16) for the justification of Lf concentrations used in this study (Lines 329-331) is not convincing because three of them are this group’s (review) articles.

2. Along the same lines, the range of physiologically relevant concentrations of Lf should be noted in this paragraph and justify the concentrations of 100-500 ug/ml used in this study.

3. What is the mechanism behind the observation that bLf is more efficient blocker than hLf? For instance, any clue from the comparison between bLf and hLf in Fig. 3, Table 2, or related molecular docking data?

4. Why is the trimetic 250kDa spike protein still intact in SDS-PAGE in Fig. 3? Neither the line 197 In Materials and Methods nor the Fig. 3 legend tell us about that.

5. Line 228 – “---heat-treated (except for Fpn)”. Why except for Fpn (how about DMT1?), any reference?

6. Like Fig. 3A, all western blots need at least positions of size markers.

7. The representative western blots in Fig. 4a look marginal expression changes.

8. This reviewer doubts if such marginal (much less than 1.5-fold) expression changes in these iron handling proteins (Fig. 4) can be convincingly and reproducibly detected by western blotting and are also physiologically important.

9. What is the mechanism through which the purified spike protein alone (not virus particle) downregulated Fpn and upregulate TfR1? (again these effects are very small, less than 1.5 fold). Briefly mention here something like the lines 610-612.

10. Lines 623-628 recapitulate the effects of the spike protein on Fpn (downregulation), which suggests increased retention of intracellular iron (into ferritin), and TfR1 upregulation. How do cells try to express more TfR1 (= more iron uptake) under already increased intracellular iron availability (via decreased Fpn expression)? Lines 652-662, seemingly too far away from the lines 623-628, do not clearly address this issue.

11. The title ” -----in different in vitro models” sounds somewhat weird (what are different?).    

Minor

1. Lines 324-329 “according to the following -----i)…iv)---were chosen following data ----” needs to be polished because it is difficult to get the author’s point.

2. The same kind of format is used from the line 403 to 407. This reviewer does not think it is helpful for readers before the presentation of the data.  

3. Actually the entire paragraph from line 322 to 338 should be extensively edited, including moving the preincubation rationale to more appropriate place. In other words, this large paragraph cannot be digested well by readers because they have not seen the data yet.

4. Line 375 – hTf was used as a control of WHAT specificity? The authors know the results but readers don’t know it yet at this point.

5. Fig. 5 is in THP-1 cells , while lines 435-436 mention different cell types.  

Reviewer 3 Report

This study uses in vitro cell models and in silico analysis to test whether lactoferrin can bind to SARS-CoV-2 spike glycoproteins to display anti-viral effects. Overall, the study was well-designed and performed. Only some minors are suggested.

NF-B pathway > Nuclear factor kappa B (NF-κB) pathway. In addition, take care of the abbreviations, such as fetal 133 bovine serum (FBS) (shown twice in the context, line 134 and line 136), TNF-α (no full name), and Transferrin Receptor 1 (TfR1), line 241 is not the first time shown in the context.

5x105 cells/ml (line 141) and 1×104 cells/well (line 150): x > ×, to be consistent.

Provide original gel blotting as supplementary data.

In addition to spike protein, is it possible that lactoferrin may bind other viral proteins encoded by SARS-CoV-2 (PMID: 35682761), can be briefly discussed.

Round 2

Reviewer 2 Report

The authors addressed all of my concerns and questions.  The revised manuscript has been much improved.